# Insight on the Interplay between Synthesis Conditions and Thermoelectric Properties of α-MgAgSb

**DOI:** 10.3390/ma12111857

**Published:** 2019-06-07

**Authors:** Julia Camut, Ignacio Barber Rodriguez, Hasbuna Kamila, Aidan Cowley, Reinhard Sottong, Eckhard Mueller, Johannes de Boor

**Affiliations:** 1Institute of Materials Research, German Aerospace Center, Linder Hoehe, 51147 Cologne, Germany; Ignacio.Barber@dlr.de (I.B.R.); Hasbuna.Kamila@dlr.de (H.K.); reinhard.sottong@web.de (R.S.); eckhard.mueller@dlr.de (E.M.); 2European Astronaut Centre, Linder Hoehe, Cologne 51147, Germany; aidan.cowley@esa.int; 3Institute of Inorganic and Analytical Chemistry, Justus Liebig University Gießen, Heinrich-Buff-Ring 17, 35392 Giessen, Germany

**Keywords:** thermoelectrics, semiconductors, synthesis optimization, MgAgSb, properties, ball-milling, sintering

## Abstract

α-MgAgSb is a very promising thermoelectric material with excellent thermoelectric properties between room temperature and 300 °C, a range where few other thermoelectric materials show good performance. Previous reports rely on a two-step ball-milling process and/or time-consuming annealing. Aiming for a faster and scalable fabrication route, herein, we investigated other potential synthesis routes and their impact on the thermoelectric properties of α-MgAgSb. We started from a gas-atomized MgAg precursor and employed ball-milling only in the final mixing step. Direct comparison of high energy ball-milling and planetary ball-milling revealed that high energy ball milling already induced formation of MgAgSb, while planetary ball milling did not. This had a strong impact on the microstructure and secondary phase fraction, resulting in superior performance of the high energy ball milling route with an attractive average thermoelectric figure of merit of $zT_{\mathrm{avg}}=$ 0.9. We also show that the formation of undesired secondary phases cannot be avoided by a modification of the sintering temperature after planetary ball milling, and discuss the influence of commonly observed secondary phases on the carrier mobility and on the thermoelectric properties of α-MgAgSb.

## 1. Introduction

With the worldwide demand for energy increasing and the environmental impact of electrical power production being considered more thoroughly, novel sources of energy are being investigated. A high portion of the primary energy consumed around the world is lost as waste heat, for example, in the cement industry (around 66%) [1] as well as in automotive applications [2]. It is well known that thermoelectric materials have the ability to convert heat directly into electricity, which makes them good candidates to turn this waste heat into a new source of energy. Thermoelectric materials have applications in various areas, from automotive to aeronautics and space applications, are currently powering the Curiosity rover on Mars and are considered fornovel heat harvesting systems for lunar energy production [3].

The maximum efficiency of a thermoelectric material is characterized by its figure of merit, $zT=\frac{\sigma S^{2}T}{\kappa }$, in which σ is the electrical conductivity, S the Seebeck coefficient and κ the thermal conductivity of the material [4]. A satisfactory of power factor ($PF=S^{2}\sigma$) is obtained with both a high Seebeck coefficient and electrical conductivity. Some thermoelectric materials are already well-optimized and commercially available, such as telluride (Bi_2_Te_3_, Sb_2_Te_3_)-based solid solutions, which represent the best thermo electric materials in the near room temperature range [5]. Considerable research has been done on high temperature materials, such as lead telluride-based nanomaterials (PbTe, GeTe), skutterudites [6,7], magnesium silicide stannide Mg_2_(Si, Sn) [8,9], half-Heusler compounds [10,11], and many more which perform best at temperatures above 350 °C. To fill the gap between those two temperature ranges, other thermoelectric materials were investigated and MgAgSb was found out to be a promising option.

In 2012, Kirkham et al. found the low-temperature phase of the p-type MgAgSb compound to be a promising thermoelectric material. The interesting phase is α-MgAgSb—it has a tetragonal structure with a distorted rock-salt lattice and is stable up to 300 °C. This phase gave an average *zT* of 0.5 between room temperature and 300 °C [12], although the material had not been optimized composition-wise and the microstructure contained impurities, which meant that the *zT* value could potentially be further improved. An effective synthesis route has been found by Zhao et al. in 2014 [13], which consists of two steps of high energy ball-milling, the first one including only pure silver and magnesium. Antimony is added to the mix in the second step of high energy ball-milling. The powder is then sintered into a pellet by hot-pressing. This method, with a slight adjustment of the stoichiometry to MgAg_0.97_Sb_0.995_, gives an average *zT* of 1.1 between room temperature and 300 °C, and the samples contain very few impurity phases. More attempts of tuning the composition and doping were made later and resulted in promising values for *zT* [14,15,16,17,18,19,20], the highest being an average of 1.2 for Ni doping on the Ag lattice site [13]. Although the material system has very promising properties, its synthesis has proven to be challenging, as evidenced by the accompanying secondary phases that are often reported. The main impurities are dyscrasite, Ag_3_Sb [12,13,15,20,21,22,23,24] and pure antimony [12,13,14,20,21,23,24], which keep appearing in most X-ray diffraction (XRD) patterns. γ-MgAgSb [15,21,22] sometimes appears as well, but not systematically. The proportions of these impurities are usually kept low with long annealing times and/or the two-step ball-milling process.

In this report, we compare directly two different synthesis methods: high energy ball milling (HEBM) (plus compaction), which has previously shown the best thermoelectric properties, and planetary ball milling (PBM) (plus compaction) which is a promising route from a technological and economical point of view [25,26]. Both of these methods are applied using a precursor powder synthesized by gas atomization, reducing the number of steps of ball-milling. We show that HEBM leads to a solid state reaction by mechanical alloying while in PBM the compounds are only fractured but no indications for a solid state reaction are found. This leads to a larger content in the secondary phase in the PBM samples and hence to inferior thermoelectric properties. We also show that the type of secondary phases can be controlled by the synthesis conditions. Some insight on the influence of these secondary phases on the electrical and thermal properties is also provided, estimating how detrimental they are with respect to the thermoelectric properties.

## 2. Materials and Methods

The two-step ball-milling process developed by Zhao et al. [13] is the one that so far has resulted in the best thermoelectric properties for the final material. Its core principle is a progressive mixing of the elements, mixing Ag and Mg first and incorporating Sb only later. The two steps of ball-milling, however, add technical complexity to the process and increase the amount of time necessary to get the final powder. For this reason, we replaced the first step of ball-milling with the synthesis of a precursor powder of MgAg with gas atomisation under argon (Hermiga Mini, PSI Ltd., Hailsham, UK). Pure elements were weighed (magnesium: Evochem, 3–6 mm pellet size, 4 N purity; silver: Sindlhauser Materials GmbH, 1–6 mm pellet size, 4 N purity) and molten in an induction melter under argon atmosphere. The melt is released through a ceramic nozzle and quenched and separated into micrometer-sized droplets by a high pressure argon gas jet [27,28]. This method has the advantage of providing large quantities of powder with each run (kg scale), and an ensured formation of MgAg during the melting (XRD pattern can be found in Appendix A in Appendix A). All samples presented in this study were synthesized from the same batch of MgAg. The precursor is then ball-milled under argon with the corresponding amount of antimony (Alfa Aesar, < 6 mm pellet size, 5 N purity), with either PBM for 12 h (Retsch PM 400, 250 rotations per minute (RPM), stop and reversal of rotation every 10 min, stainless steel jar and balls, powder to ball ratio PBR 10:3, dry) or HEBM for 5 h (SPEX 8000 D mill, 900 RPM, no stopping period, stainless steel jar and balls, PBR 16:1). A nominal stoichiometry of MgAg_0.97_Sb_0.995_ was aimed for, as this is the composition tested by Liu et al. which gave the best properties [14]. The samples were sintered using current assisted direct sintering pressing (Dr Fritsch DSP510 SA) for 5 minutes at 300 °C and 90 MPa under vacuum, except if mentioned otherwise in the following parts. The sintered pellets were then annealed for 3 h at 250 °C to ensure the transition of any potential β-MgAgSb back to α-MgAgSb.

As for characterization, the electrical conductivity and Seebeck coefficient of the samples were measured with an in-house device with a four probe technique [29,30], the thermal diffusivity α by a laser flash technique with a NETZSCH LFA 427 apparatus (NETZSCH-Gerätebau GmbH, Selb, Germany) or with a XFA 467HT Hyperflash apparatus (NETZSCH-Gerätebau GmbH, Selb, Germany). The thermal conductivity (κ) was calculated from $\kappa =\alpha \rho C_{p}$, where ρ and $C_{p}$ are sample density and heat capacity depending on the composition at constant pressure, respectively. ρ was measured for each sample using Archimedes’ method. The *C_p_* value was calculated using the Dulong–Petit limit, which states that $C_{v}$ only depends on the number of atoms in the body and can be approximated by 3 *R*, with *R* being the ideal gas constant [31]. Normally, *C*_p_ varies with temperature and should be approximated by $C_{p}=C_{v}^{\mathrm{DP}}+\frac{9E_{T}^{2}T}{\beta_{T}\rho }$, where $C_{v}^{\mathrm{DP}}\;$ is the Dulong–Petit limit, and $E_{T}$ and $\beta_{T}$ respectively are the linear coefficient of thermal expansion and isothermal compressibility dependent on composition [32]. However, reproducible mechanical data for MgAgSb is not currently available in the literature as only a few studies have been reporting it [15,33], and Zhao et al. and Shuai et al. reported differential scanning calorimetry (DSC) measurements which indicated that the $C_{p}$ was reasonably stable in the range of 300–600 K [13,17], which was the range we worked within. As a consequence, *C_p_* was approximated by the Dulong–Petit limit to calculate the thermal conductivity. We specified the uncertainties for *S*, *σ* and *κ*, for the different samples as ± 5%, ± 5% and ± 8% based on a comparison with the National Institute of Standards and Technology (NIST) low temperature standard for the Seebeck coefficient [34] and internal reference measurements on a high temperature standard [35]. Our estimates were comparable to the numbers obtained in an international round robin test.

A Siemens D5000 Bragg–Brentano diffractometer (Bruker Corporation, Billerica, MA, USA) with a secondary monochromator and a D8 Advance A25 Twin/Twin Setup (Bruker) were used for XRD analysis of powders and pellets. The specifics used in the Siemens system were Cu-K_α_ radiation (1.5406 Å) in the range (2θ: 20°–80°) with a step size of 0.01°. For the Bruker system, the specifics used in the system were Cu-K_α_ radiation in the range (2θ: 20°–120°) with a step size of 0.01°. Scanning electron microscopy (SEM) and energy dispersive X-ray (EDX) was performed with a Zeiss Ultra 55 SEM.

## 3. Results

### 3.1. Comparison of High Energy Ball-Milling and Planetary Ball-Milling

Figure 1 displays the XRD patterns of the PBM and HEBM powders and the sintered pellets. The pellet made from the PBM powder was pressed at 85 MPa. It can be seen that PBM did not initiate MgAgSb formation to a significant extent in the powder as there were well resolved peaks of MgAg and pure antimony, while in 5 hours of HEBM, the compounds started to react to MgAgSb, and on the 10 h pattern we can clearly see α-MgAgSb main peaks, indicating this to be the highest content phase. It is however noteworthy that some dyscrasite (Ag_3_Sb) began to form after 10 hours of high energy ball-milling, which was the reason why this powder was not used further to sinter pellets.

The difference between the powders caused significant differences in sintered samples, as the sample synthesized by PBM contained various impurities (Mg_3_Sb_2_, MgAg, Sb, Ag_3_Sb and Ag_3_Mg) while the one using HEBM was almost pure and contained only a small proportion of Mg_3_Sb_2_. The phase fractions of the sintered samples obtained by Rietveld refinement are reported in Table 1. The main impurity phase in the PBM sample was pure antimony, which represented more than 16% in weight. It should be noted that the quantitative XRD phase analysis had some uncertainties for this system, as a lot of secondary phases had very closely located main diffraction peaks. XRD patterns of common secondary phases already found in samples, or which could be found according to the phase diagram, are plotted in Appendix A in the Appendix A. It can be seen that pure silver and Ag_3_Mg had similar patterns, and that they shared their main peak with γ-MgAgSb and AgMg, which can only be distinguished by their lower intensity peaks. Even if the main peaks were separated by at least 0.04° (three measurement points), a slight difference in stoichiometry could shift peaks, so the risk of not distinguishing these phases did exist, especially if the impurities were in low proportion (only the highest intensity peak was visible). The main peak of dyscrasite was located farther from the others. Pure antimony and Mg_3_Sb_2_ were easily identifiable and could be safely distinguished from other secondary phases.

Figure 2 shows the observed microstructure of the sintered samples. It can be seen that PBM led to the formation of structures made of concentric rings, with unreacted MgAg grains at the center, surrounded by Ag_3_Mg and then Mg_3_Sb_2_. Wide areas of pure antimony could also be seen. Ag_3_Mg and MgAg were, however, not detected in XRD refinement. As those phases had their main peak at a similar angle to the one of γ-MgAgSb (at around 38°), it was possible that they were coincident with it in the XRD pattern.

The sample synthesized with HEBM contained some impurities as well, but in a much smaller quantity (Table 1). The ring structures were still present, but smaller, and some precipitates of dyscrasite could also be seen. It is seen in Table 1 that the XRD pattern showed Mg_3_Sb_2_, but no peak of MgAg, Ag_3_Mg nor dyscrasite were fitted in the pattern. It was possible that their amount was so low that XRD could not detect them.

Thermoelectric properties of both pellets are reported in Figure 3. The lattice thermal conductivity (which also includes the bipolar contribution to thermal conductivity) was obtained by the equation $\kappa_{\mathrm{lat}}+\kappa_{\mathrm{bip}}=\kappa_{\exp}-\kappa_{e}=\kappa_{\exp}-LT\sigma$ where the Lorenz number is given by $L=1.5+\exp\left(-\frac{\left|S\right|}{116}\right)$ [36].

Overall, HEBM led to properties that were closer to the best literature report, with a maximum zT of roughly 1.1. PBM gave high electrical conductivity but a low Seebeck coefficient in comparison, reducing the power factor, and had higher thermal conductivity despite a lower lattice thermal conductivity, reducing the *zT* to an average of 0.4. It was possible to estimate the carrier concentration and the carrier mobility using the single parabolic band model (SPB) [32]. In this model SPB, the transport properties are interrelated by a set of relatively simple equations given by
(1)$$S=\frac{k_{B}}{e}\times \left(\frac{2F_{1}\left(\eta \right)}{F_{0}\left(\eta \right)}-\eta \right)$$
(2)$$p=4\pi {\left(\frac{2m_{d}^{*}k_{B}T}{h^{2}}\right)}^{1.5}F_{\frac{1}{2}}\left(\eta \right)$$
(3)$$\eta =\frac{E_{F}}{k_{B}T}$$
where η is the reduced chemical potential, $m_{d}^{*}\;$ is the density of states effective mass, *p* is the carrier (holes) concentration, $k_{B}$ is the Boltzmann’s constant and $F_{i}\left(\eta \right)$ is the Fermi integral of order i.

We used the relation between *S* and *p* (Pisarenko plot), employing an effective mass of $2.7\;m_{0}$ as reported in the literature to deduce the carrier concentration of the samples from our measured *S* data [14]. The mobility μ was then calculated using the following equation:(4)σ=peµ

Results of the calculations are displayed in Table 2. Literature values of carrier concentration and mobility were not calculated with the same method but directly taken from Liu et al.’s publication, where they obtained their values with Hall measurements. It was seen that both samples exhibited mobility inferior to the best comparable literature report, roughly by a factor of 2 for HEBM, and a factor of 4 for PBM, which was probably due to the difference in the fraction of impurity phases in the samples. The sample by PBM also exhibited a much higher (and thus farther from the presumable optimum) carrier concentration, while the HEBM sample had a carrier concentration similar to the benchmark sample, which partially explained why its properties were closer to the reference.

### 3.2. Influence of Sintering Temperature with Planetary Ball-Milling

Since planetary ball-milling has the benefits of economically viable scalability, optimization of the sintering step after ball milling for enhanced properties is highly desirable. In order to understand the mechanisms at play during sintering after planetary ball-milling and possibly improve the results obtained with this process, the sintering temperature was changed between 300, 350 and 400 °C. All these samples were pressed at 85 MPa instead of 90 MPa. Indicative SEM images of these samples are shown in Figure 4. It is seen that at and above 350 °C, the concentric ring structures disappeared, leaving only pure antimony in the matrix. Figure 5 shows the XRD patterns of the samples and Table 3, the phase proportions in the samples obtained by Rietveld refinement of these patterns. The content of pure antimony was higher for 400 °C than for 350 °C (11.4 wt.% and 7.9 wt.%, respectively). We also found a small amount of dyscrasite, Ag_3_Sb, in the samples sintered at 300 °C, in addition to the other secondary phases composing the ring structures. This was not surprising, as dyscrasite is a common impurity in this system and has already been found in samples of several studies [12,13,15,20]. Some amount of γ-MgAgSb was also fitted in the samples pressed at 350 °C and 400 °C. This phase was supposed to only form above 360 °C, but with an experimental error [37] it was possible that the sample got heated at this temperature or above, rather than at 350 °C precisely.

In Figure 6, the thermoelectric properties of the PBM samples with different sintering temperatures are displayed. Carrier concentration and mobility at room temperature were calculated in Table 4 to help for the analysis of the effect of the impurities on the thermoelectric properties.

Compared to the literature, the sample pressed at 300 °C showed a slightly lower lattice thermal conductivity, had a significantly higher carrier concentration and its mobility was inferior to literature by a factor of 3. The samples pressed at 350 and 400 °C also had a very high carrier concentration, their lattice thermal conductivity was higher and their mobility was still inferior to the literature by a factor of 2. We observed a decrease in carrier concentration and an increase in carrier mobility when increasing from 300 to 350 °C; 350 and 400 °C samples showed similar values. We also observed that the reduced amount of (metallic or semimetallic) secondary phases (mainly MgAg and Mg_3_Sb_2_) for the samples pressed at higher temperature correlated with a reduction of the carrier concentration. This translated into a significant increase in the Seebeck coefficient without a reduction of the electrical conductivity, which greatly improved the power factor for the higher sintering temperatures.

However, this was compensated by the higher lattice thermal conductivity so that the samples sintered at 350/400°C did not show a significantly improved *zT* compared with the sample sintered at 300 °C. Compared with the reference sample, all samples had a high carrier concentration, which partially explains their poor performance.

We performed further studies on the planetary ball-milling process parameters (influence of sintering time and stoichiometry), but in all studies, the thermoelectric properties were inferior to literature reports, and our own HEBM results and the microstructure was complex and multi-phase. We conclude that a planetary ball-milling only approach is not suitable for this system, as it does not induce the mechanical alloying which makes the sintering step easier. Other attempts of planetary ball-milling processes have also led to *zT* inferior to high energy ball-milling processes (0.7) and low mobility without getting a phase-pure micro-structure either [20].

## 4. Discussion

### 4.1. Effect of Synthesis Route on Microstructure and Thermoelectric Properties

The concentric ring structure obtained with planetary ball-milling, and on a smaller scale with high energy ball-milling, was presumably an intermediate step of the reaction between antimony and MgAg. This would indicate that before ultimately forming MgAgSb, the magnesium contained in MgAg diffused partially out and reacted with antimony, forming a Mg–Sb region (probably Mg_3_Sb_2_) on the outside border of the initial MgAg grain, and leaving a Mg-poor region behind (Ag_3_Mg). This hypothesis was supported by the fact that the concentric layers disappeared when the sintering temperature increased and, as a consequence, the kinetics of the interdiffusion and homogenization for formation of the target phase increased. It is also explained by the fact that PBM does not initiate mechanical alloying in the powder, while HEBM does. It is plausible that the (further) formation of MgAgSb during the sintering step was facilitated by these seeds that had already reacted, which explains the large MgAgSb fraction obtained of HEBM compared with PBM. Increasing the sintering temperature when there was already only pure antimony left did not contribute to dissolve it; on the contrary, the proportion of pure antimony in the microstructure increased. The observed increase in Sb-content above 350 °C could indicate a decreasing solubility of Sb, but further research is required on this, as a no phase diagram for temperatures below 450 °C is available [38].

Despite the high *zT* obtained for the HEBM sample, the sample properties could still be further optimized to reach the best values reported in the literature. Our samples had a non-optimized carrier concentration, which was too high for PBM samples and slightly too low for the HEBM sample. In first principle calculations, Sheng et al. calculated the relation between *zT* and carrier concentration, which showed that the optimal carrier concentration would be between 5 and 7 × 10^19^ cm^−3^ (between room temperature and 550 K, respectively) [39]. The value of 5.5 × 10^19^ cm^−3^ for our reference in Liu et al.’s study is in accordance to these calculations and shows the highest *zT* for undoped α-MgAgSb. The carrier concentration of our PBM samples was between 20 and 30 × 10^19^ cm^−3^, which was far away from the optimal range and explains the poor properties. Further optimization of the thermoelectric transport properties could thus be achieved by carrier concentration optimization. This is feasible by subtle variation of the matrix composition, in particular the Sb content [14]. This is particularly promising if the amount of secondary phases can be minimized.

It is clear that the presence of impurities in the samples made by PBM were also altering the thermoelectric properties of the material. The sample containing the big concentric structures obtained with PBM at 300 °C showed a decrease in carrier mobility, which could be imputed to the numerous interfaces between the different phases. Furthermore, this could also be due to the difference in composition of the matrix in Ag or Sb content, which had an impact on the defect concentration and thus on the carrier concentration. The disappearance of the concentric rings with a higher sintering temperature increased the carrier mobility. However, these improvements did not allow an increase of the *zT* by a significant extent, as the samples still had a high carrier concentration, which could be optimized in further studies, and a mobility that was still low compared with the literature due to secondary phases and an increased lattice thermal conductivity, probably due to the impurities. Other studies have also shown a low mobility in samples made with PBM, even without featuring such ring structures [20].

While the variation of PBM parameters did not significantly improve the sample quality and thermoelectric properties, employing HEBM improved things considerably. The sample sintered from HEBM powder at 300 °C had the biggest α-MgAgSb proportion of all samples in this study, and its carrier concentration was very close to the reported optimum value, although its mobility was half the reference. Presumably this difference in mobility decreases with temperature, as the importance of secondary phases scattering on carrier mobility usually decreases with temperature [40].

With the carrier concentration closer to the optimized value, the properties were closer to the literature, with a similar Seebeck coefficient, lower electrical and thermal conductivity and an average *zT* of 0.9. The explanation for the differences compared with PBM was that mechanical alloying seemed to be more effective to increase the proportion of matrix in the microstructure than in increasing the sintering temperature. It was possible that the reaction between the compounds needed a higher temperature to be completed more rapidly, but in order to avoid the phase transition with γ-MgAgSb, the sintering temperature had to be limited to 360 °C, and hence mechanical alloying remained then the best option to complete the reaction.

### 4.2. Effect of Impurity Phases on Thermoelectric Properties

As the synthesis of MgAgSb is often accompanied by undesired secondary phases [12,13,14,15,20,21,22,23,24], it is important to understand their influence on the thermoelectric properties. Correlations between thermoelectric properties and the content of Ag_3_Sb (dyscrasite), pure Sb and γ-MgAgSb are discussed below, while that of Mg_3_Sb_2_ could not be separated.

There is some uncertainty on these analyses coming from small differences in the proportions of other impurities than the one that is being isolated in the compared samples (<2%wt). The results of the Rietveld refinement are also a source for some error.

Figure 7 shows lattice thermal conductivity and *zT* for various Ag_3_Sb contents and Table 5 shows the calculated carrier concentration and mobility for the same samples. Other thermoelectric properties and SEM investigations are available in Appendix A in the Appendix A. Figure 7 shows a decrease in the overall and lattice thermal conductivities with increasing dyscrasite content. In Table 5 it is seen that both carrier concentration and mobility increased with increasing dyscrasite content. This is in disagreement with what has been found by Lei et al. [20], who states that the presence of Ag_3_Sb reduces the hole carrier concentration, but it is in agreement with the results of Zheng et al. [25], who also observed an increase in carrier concentration with increasing impurity content. This increase in carrier concentration can be explained by the metallic nature of dyscrasite [41].

The decrease in lattice thermal conductivity with higher dyscrasite content can be explained by an increase of grain boundaries scattering and/or defect scattering. Liu et al. found a contradicting influence of the impurity content on the lattice thermal conductivity [15], which may have been due to the presence of a second impurity instead (Ag_3_Li_10_), as our results are in agreement with a publication from Zheng et al. [25] who explain the decrease in lattice thermal conductivity by the combination of defect scattering and the grain boundaries scattering. The observed increase in hole mobility is somewhat surprising—it could partially be explained by an effective medium effect due to the metallic nature of dyscrasite. The increase of carrier concentration and mobility and the decrease in lattice thermal conductivity resulted in an increase in zT, which shows that dyscrasite might not have a strong detrimental effect.

With increasing γ-MgAgSb content, it is seen in Table 6 that both the carrier concentration and carrier mobility decreased. Figure 8 shows the thermoelectric properties, it is seen that with increasing γ-MgAgSb content, the Seebeck coefficient increased since the carrier concentration decreased, the electrical conductivity decreased and the lattice thermal conductivity increased. The *zT* was reduced even if the carrier concentration got closer to the optimized value from the literature, due to the increase in lattice thermal conductivity and the decrease in mobility. Like for dyscrasite, the evolution of mobility was probably mostly related to the difference in grain boundaries scattering. The nature of γ-MgAgSb being unclear (metallic or narrow band gap semiconductor depending on calculations [42]), it is difficult to explain the evolution we see in carrier concentration. This impurity has not been found much in α-MgAgSb publications and the phase has not been studied on the temperature range below 300 °C. Thus we can hardly conclude on whether the detrimental effects we see in the properties were the result of the presence of γ-MgAgSb or rather of its effect on the un-optimized composition of the matrix material. We can, however, reasonably assume that it was a combination of both effects.

## 5. Conclusions

We have implemented a modified route for the synthesis of MgAgSb—instead of starting a ball-milling synthesis route from the elements, we started with MgAg synthesized by an upscaled gas atomization process. This was followed by a ball-milling step with Sb, where we compared directly planetary ball-milling and high energy ball-milling. While planetary ball-milling resulted basically in a crushing of the reactants, high energy ball-milling clearly induced the formation of MgAgSb during the milling. This resulted in samples with much less secondary phase content and, hence, attractive thermoelectric properties—$zT_{max}\approx 1.2\pm 0.17$ and an average thermoelectric figure of merit of 0.9. We can also show that a systematic variation of the sintering temperature for the planetary ball milling did not result in satisfactory results—low sintering temperatures resulted in an incomplete reaction, while high sintering temperatures resulted in the formation of the undesired and persistent γ-MgAgSb, always resulting in a poor thermoelectric performance. We thus conclude that high energy ball-milling is clearly superior to planetary ball-milling for the synthesis of MgAgSb. Finally, we also investigated the influence of different secondary phases on the thermoelectric properties and found that the influence of γ-MgAgSb is presumably detrimental while that of dyscrasite is relatively mild.

## Figures and Tables

**Figure 1 materials-12-01857-f001:**
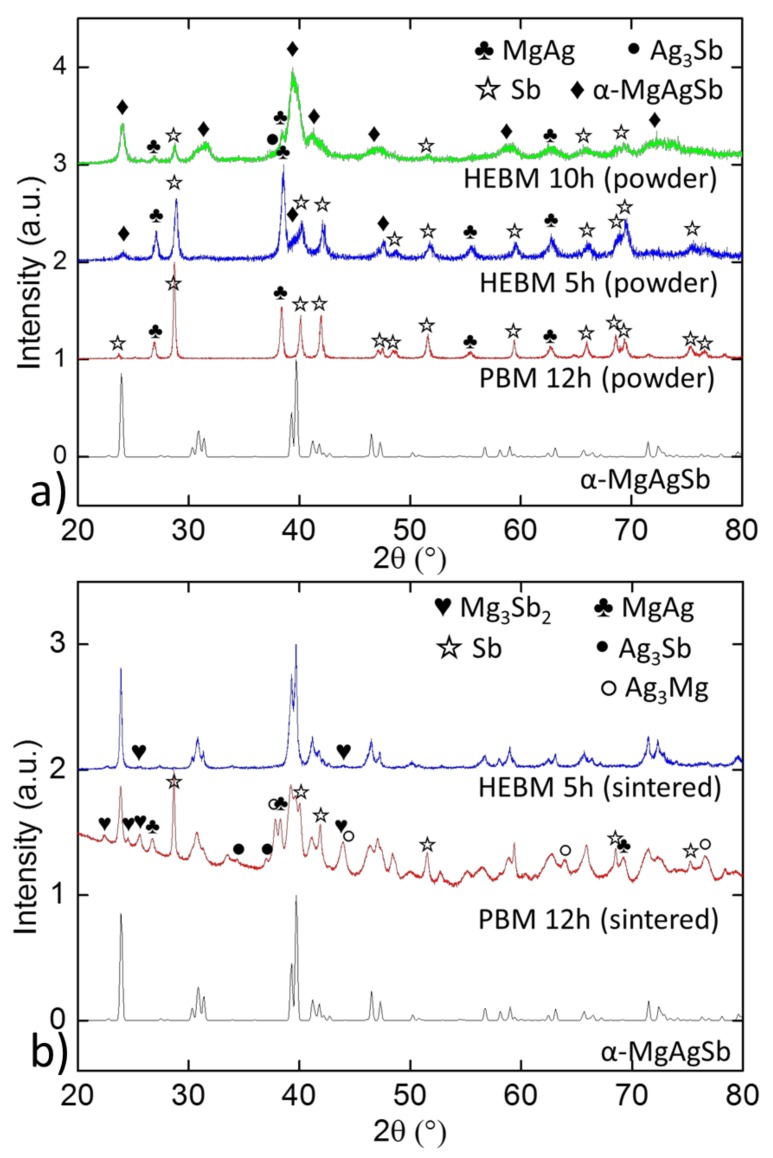
XRD patterns of powders (**a**) and sintered samples (**b**) comparing high energy ball-milling with planetary ball-milling and the theoretical pattern of α-MgAgSb [12].

**Figure 2 materials-12-01857-f002:**
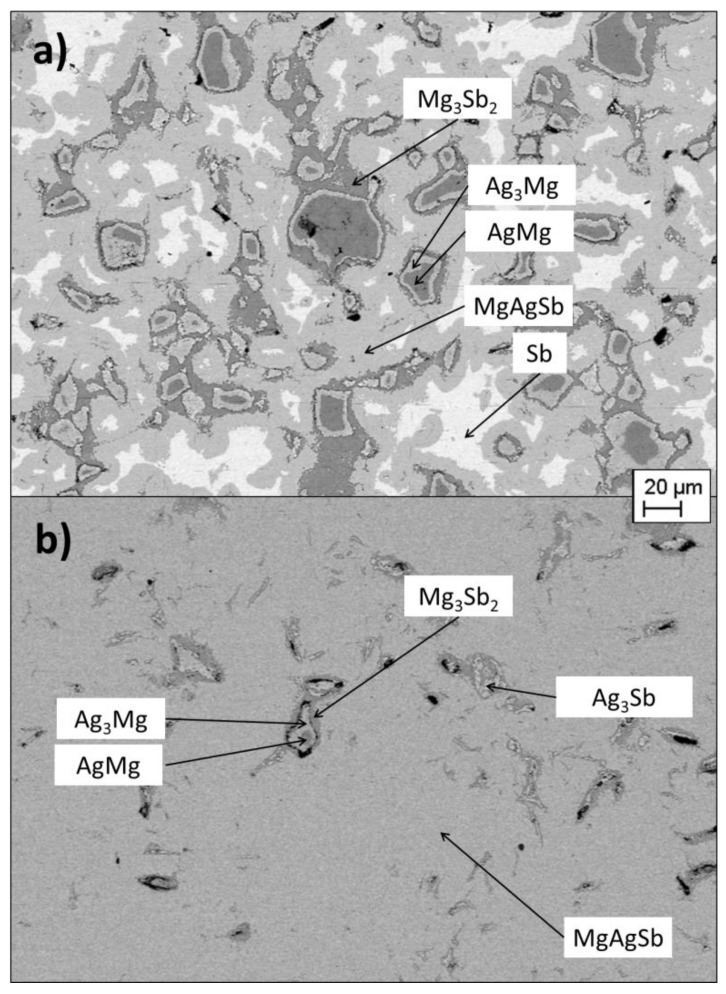
SEM images of sintered samples from planetary ball-milling (**a**) and high energy ball-milling (**b**).

**Figure 3 materials-12-01857-f003:**
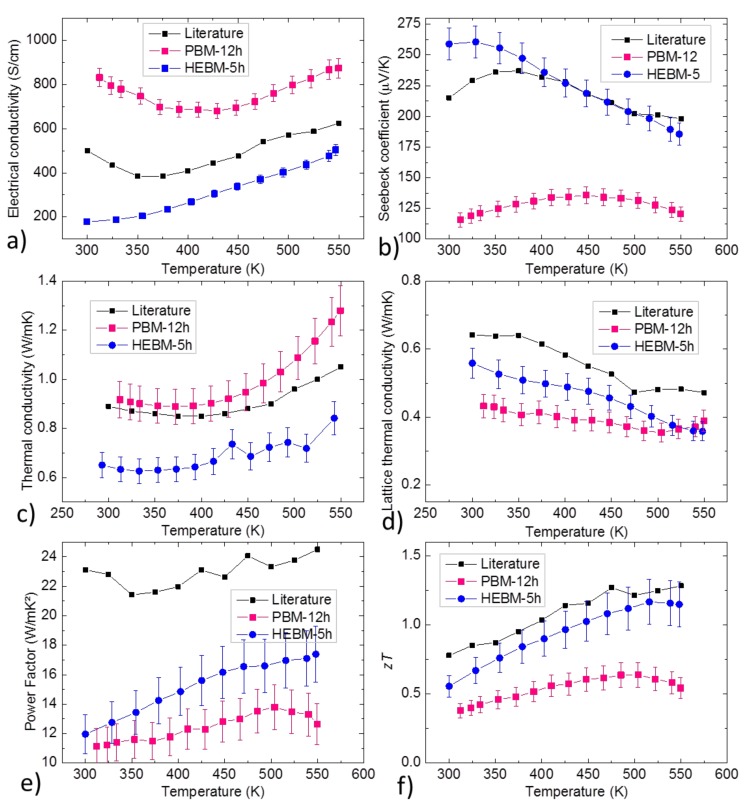
Temperature dependent (**a**) electrical conductivity, (**b**) Seebeck coefficient, (**c**) thermal conductivity, (**d**) lattice thermal conductivity, (**e**) power factor and (**f**) figure of merit of sintered PBM and HEBM samples. Literature values are taken from Liu et al. [14].

**Figure 4 materials-12-01857-f004:**
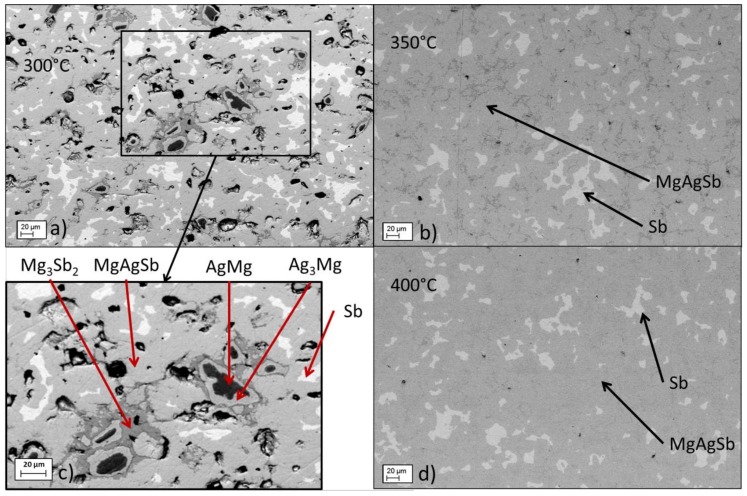
SEM images of the effect of sintering temperature on the microstructure after planetary ball-milling: (**a**) 300 °C, (**b**) 350 °C, (**c**) 300 °C zoomed, (**d**) 400 °C.

**Figure 5 materials-12-01857-f005:**
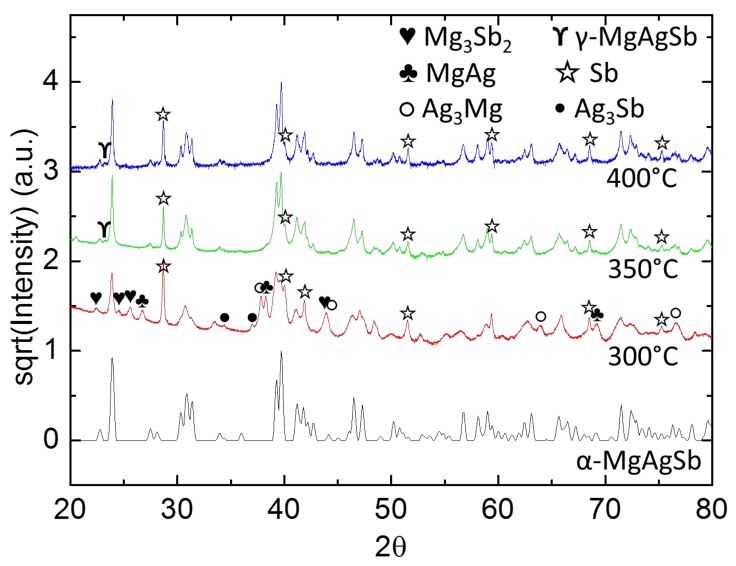
XRD investigation of samples with differing sintering temperatures after planetary ball-milling.

**Figure 6 materials-12-01857-f006:**
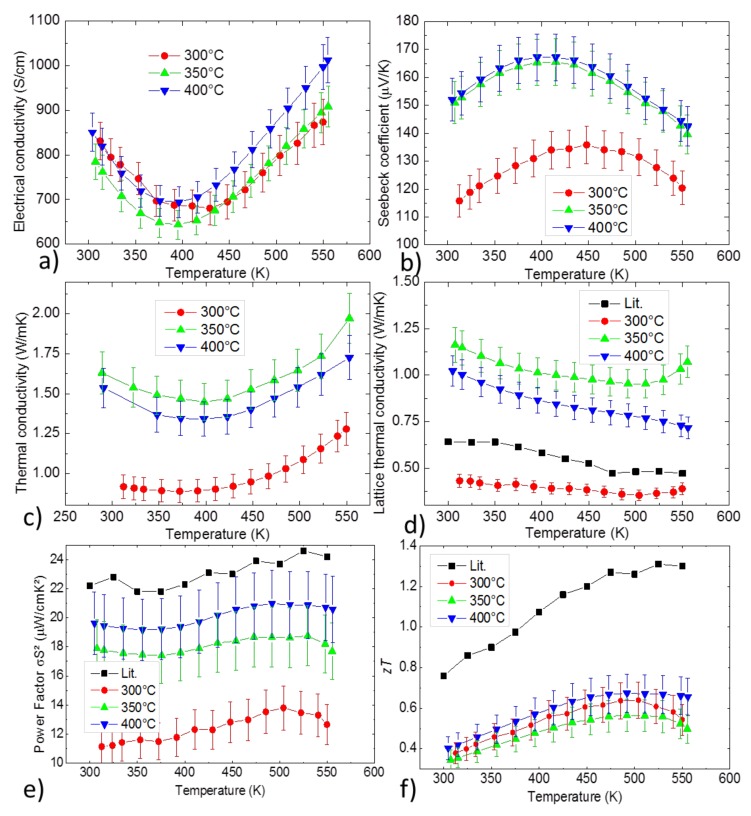
Temperature dependent (**a**) electrical conductivity, (**b**) Seebeck coefficient, (**c**) thermal conductivity, (**d**) lattice thermal conductivity, (**e**) power factor and (**f**) figure of merit of samples from planetary ball milling with different sintering temperatures.

**Figure 7 materials-12-01857-f007:**
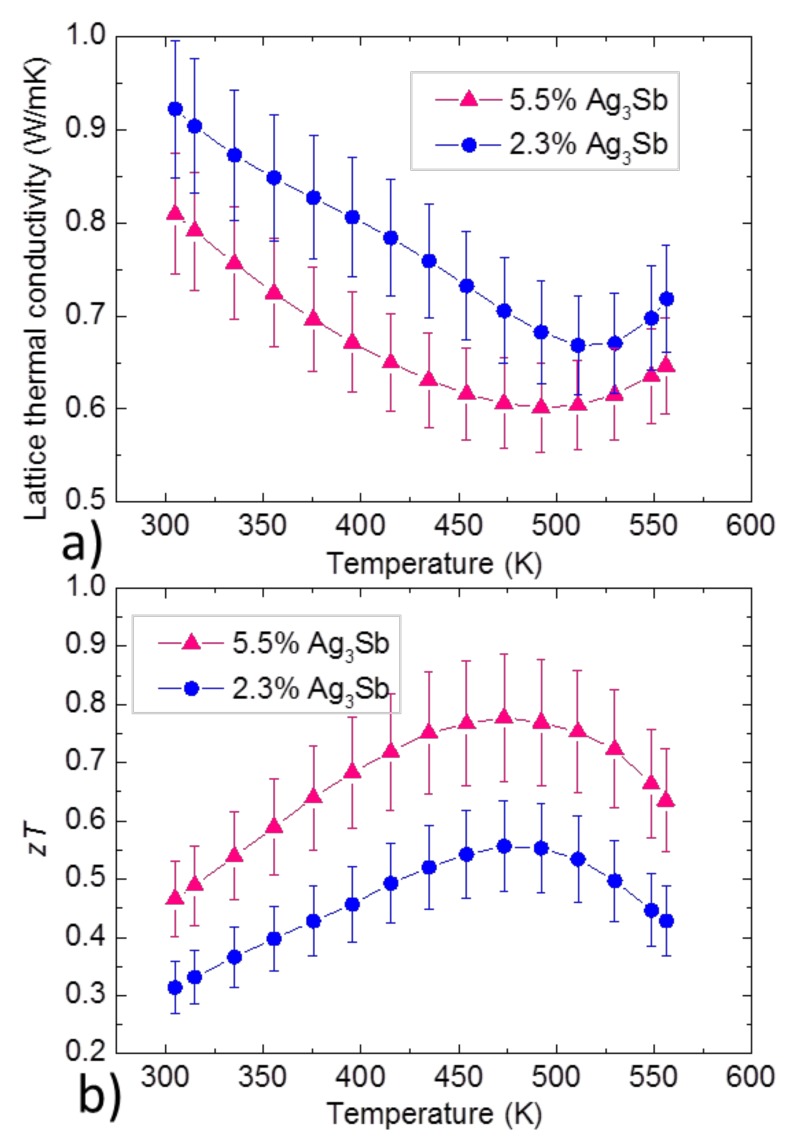
Comparison of (**a**) lattice thermal conductivity and (**b**) figure of merit for different dyscrasite contents. The samples were made by PBM.

**Figure 8 materials-12-01857-f008:**
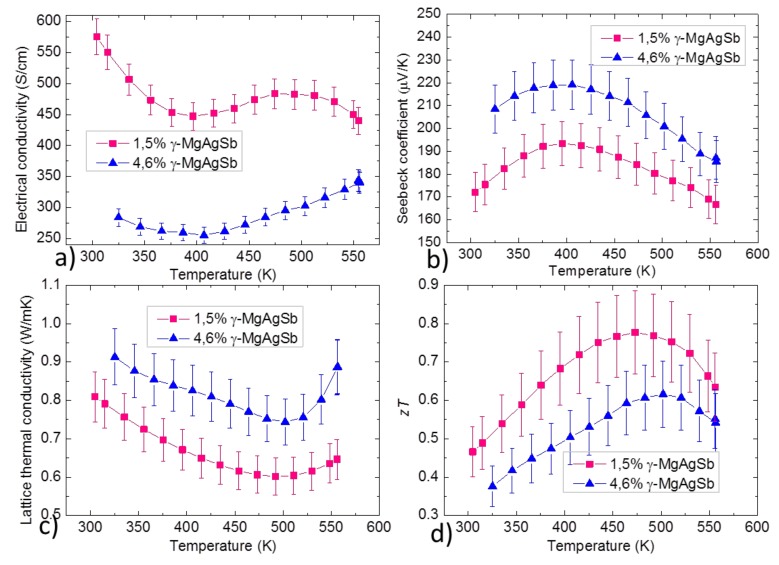
Comparison of temperature dependent (**a**) electrical conductivity, (**b**) Seebeck coefficient, (**c**) lattice thermal conductivity and (**d**) figure of merit for different γ-MgAgSb contents. Both samples were made with PBM.

**Table 1 materials-12-01857-t001:** Phase proportions in the samples obtained by Rietveld refinement for different ball-milling types. GOF = goodness of fit, Rwp = weighted profile R factor, PBM = planetary ball milling, HEBM = high energy ball milling.

Milling Type	α-MgAgSb (wt.%)	Dyscrasite (Ag3Sb) (wt.%)	Sb (wt.%)	Mg3Sb2 (wt.%)	Ag3Mg (wt.%)	AgMg (wt.%)	GOF	Rwp
PBM	60.9	2.2	16.8	5.7	0.6	13.7	5.54	9.41
HEBM	98.4	0	0	1.6	0	0	2.27	20.64

**Table 2 materials-12-01857-t002:** Estimated carrier concentration and mobility at room temperature of the pellets synthesized with different ball-milling methods.

Milling Type	Seebeck Coefficient (µV/K)	Electrical Conductivity (S/cm)	p (cm^−3^)	µ (cm^2^/Vs)
Liu et al.	215	500	5.5 × 10^19^	55
PBM	114	860	3.3 × 10^20^	16
HEBM	259	178	4.5 × 10^19^	25

**Table 3 materials-12-01857-t003:** Phase proportions in the samples obtained by Rietveld refinement for the study on sintering temperature.

**Sintering Temperature** **(°C)**	**α-MgAgSb** **(wt.%)**	**Dyscrasite (Ag_3_Sb)** **(wt.%)**	**Sb** **(wt.%)**	**Mg_3_Sb_2_** **(wt.%)**	
300	60.9	2.2	16.8	5.7	
350	89.9	0	8.8	0	
400	86.7	1.8	11.4	0	
**Sintering Temperature** **(°C)**	**Ag_3_Mg** **(wt.%)**	**AgMg** **(wt.%)**	**γ-MgAgSb** **(wt.%)**	**GOF**	**Rwp**
300	0.6	13.7	0	5.54	9.41
350	0	0	1.4	5.76	13.37
400	0	0	0.1	2.92	19.32

**Table 4 materials-12-01857-t004:** Carrier concentration and mobility calculations at room temperature for the sintering temperature study.

Sintering Temperature (°C)	Seebeck Coefficient (µV/K)	Electrical Conductivity (S/cm)	p (cm^−3^)	µ (cm^2^/Vs)
Liu et al. (300 °C)	215	500	5.5 × 10^19^	55
300	114	860	3.3 × 10^20^	16
350	153	762	1.8 × 10^20^	26
400	154	817	1.8 × 10^20^	28

**Table 5 materials-12-01857-t005:** Calculated carrier concentration and mobility at room temperature for different dyscrasite contents.

Ag_3_Sb (wt.%)	Seebeck (µV/K)	Electrical Conductivity (S/cm)	p (cm^−3^)	µ (cm^2^/Vs)
0 (Liu et al.)	215	500	5.50 × 10^19^	55
2.3	206	255	8.8 × 10^19^	18
5.5	172	574	1.4 × 10^20^	26

**Table 6 materials-12-01857-t006:** Calculations of carrier concentration and mobility at room temperature for different γ-MgAgSb contents.

γ-MgAgSb (wt.%)	Seebeck (µV/K)	Sigma (S/cm)	p (cm^−3^)	µ (cm^2^/Vs)
0 (Liu et al.)	215	500	5.5 × 10^19^	55
1.5	172	574	1.4 × 10^20^	26
4.6	202	320	9.3 × 10^19^	21

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
