# Peer review of "Insight on the Interplay between Synthesis Conditions and Thermoelectric Properties of α-MgAgSb"

_materials, 2019, doi:10.3390/ma12111857_

Round 1

Reviewer 1 Report

The manuscript reports on the thermoelectric (TE) performance of MgAgSb compounds that were synthesized by different mechano-chemical routes. The authors conclude that the samples prepared by high energy ball-milling were found to exhibit better TE properties than planetary ball-milled samples. Though the reported TE properties are not that attractive and lower than what was already existing in the literature, the authors have made sincere attempts to understand the different processes and the role of contribution of constituent phases to the overall properties. The rationale for the work is clearly defined and the manuscript is well presented. Hence, it can be considered for publication in Materials after a minor revision.

Few minor comments and queries:

1.     Page 3 – The sintered pellets were annealed to ensure a desirable phase (α-MgAgSb). Did the authors check if they get some β-MgAgSb phase without annealing? What about the TE properties of the non-annealed samples? Is there any significant variation?

2.     Page 7 – The authors have computed the carrier concentration values from the Pisarenko relation by employing a constant effective mass. Though it’s an accepted way of estimating carrier concentration, it not so precise. Why didn’t they instead try Hall measurement to estimate the carrier concentration?

3.     Page 11 – The authors have reported that the zT of their HEBM sample could be further improved to reach the values as reported in the literature. But they haven’t given any perspectives on how to improve the zT or how to optimize the carrier concentration in their mechanically milled MgAgSb compounds.

4.     Page 2 – In the introductory section, when mentioning about the state-of-the-art materials such as PbTe and GeTe, the authors can consider adding few more recent publications such as: Inorg. Chem. 2018, 57, 12976–12986.

5.     Error codes for references are found throughout the manuscript. Rectify those error lines.

Author Response

1.       "Page 3 – The sintered pellets were annealed to ensure a desirable phase (α-MgAgSb). Did the authors check if they get some β-MgAgSb phase without annealing? What about the TE properties of the non-annealed samples? Is there any significant variation?"

We did not perform XRD on non-annealed samples. Measurements of the TE properties on non-annealed samples were performed at the initial stages of establishing the synthesis route (but not on the samples discussed in the manuscript). We found that annealing improves the thermal stability of the samples, i.e. the difference between heating and cooling curves and followed that procedure in the following.

2.        "Page 7 – The authors have computed the carrier concentration values from the Pisarenko relation by employing a constant effective mass. Though it’s an accepted way of estimating carrier concentration, it not so precise. Why didn’t they instead try Hall measurement to estimate the carrier concentration?"

The Hall facility was not available at the time the samples were produced (shut down of the power supply for the magnet). Since then, some samples broke or have been cut for further investigations. Our Hall facility (similar to the one described in [1]) requires circular samples so we could not have obtained measured Hall data for all the samples. In order to avoid inconsistencies we relied on the Pisarenko-relation instead.  

[1] Borup, K.A., et al., Measurement of the electrical resistivity and Hall coefficient at high temperatures. REVIEW OF SCIENTIFIC INSTRUMENTS, 2012. 83(12).

3.       "Page 11 – The authors have reported that the zT of their HEBM sample could be further improved to reach the values as reported in the literature. But they haven’t given any perspectives on how to improve the zT or how to optimize the carrier concentration in their mechanically milled MgAgSb compounds."

Previous reports show that the carrier concentration can be adjusted by subtle variation of the matrix composition, in particular the Sb content. This makes particular sense if the samples are secondary-phase free which we haven’t been able to achieve so far. Given the constant presence of secondary phases, we suspect we have magnesium evaporation, eventually leading us outside of the single-phase area of the phase diagram, resulting in a nonoptimized matrix composition and carrier concentration.To clarify this point in the manuscript we have added:

“Further optimization of the thermoelectric transport properties can thus be achieved by carrier concentration optimization. This is feasible by subtle variation of the matrix composition, in particular the Sb content []. This is particularly promising if the amount of secondary phases can be minimized.”

At line 304.

Points 5 (dynamic link) has been corrected in the manuscript. Point 4 (reference to PbTe and GeTe) will be added in the later weeks as an Endnote dynamic link, as I did not have the necessary software with me and I could not have colleagues adding the reference for me in time.

Reviewer 2 Report

Authors report here the comparative study of thermoelectric properties of α-MgAgSb synthesized by different processes and under varying conditions. This is an interesting study and worthwhile for publications. There are some minor queries.

1. What are the densities of the samples grown by different processes under different conditions?

2. In Table 4 the carrier concentration decreases with the increase in sintering temperature. Is there any explanation behind this?

3.  In page 13, Line 349-350 “Both the increase in mobility and decrease in lattice thermal conductivity with higher dyscrasite content could be explained by a decrease of grain boundaries scattering.” This statement is confusing; with the increase in dyscrasite content the number of grains is supposed to increase, enhancing the grain boundary scattering.

4.   In page 2, line 93 “Error! Reference source not found”

5.    In page 13, line 341 “Error! Reference source not found”

Author Response

       "What are the densities of the samples grown by different processes under different conditions?"

The sample synthesized with HEBM has a density of 92% of the theoretical density. The samples synthesized with PBM have an increasing density with increasing sintering temperature (87, 97, 95% respectively for the samples pressed at 300, 350 and 400°C). This is explained by the increasing softness of the material with temperature, which improves the compaction. The HEBM samples has a higher density than the matching sample made with PBM, this could be explained by a more homogeneous powder (size, shape, brittleness/ductility).

2.       "In Table 4 the carrier concentration decreases with the increase in sintering temperature. Is there any explanation behind this?"

The 300°C sample has higher p, 350 and 400 °C have similar p. This is in agreement with the observed microstructure of the samples: the 300°C has mostly metallic secondary phases (ring structure) while 350-400°C samples only have Sb. We also note that the secondary phase content (total and metallic) is smaller for the 350 and 400°C. It si plausible that the metallic secondary phases are partially responsible for the change in carrier concentration. However, as there is a complex interplay between carrier concentration and matrix composition [1] and potential secondary phases, the detailed mechanism remains unclear. We have added that thought into the manuscript at l 262:

“We also observe that the reduced amount of (metallic or semimetallic) secondary phases (mainly MgAg and Mg3Sb2) for the samples pressed at higher temperature correlates with a reduction of the carrier concentration.”

3.       "In page 13, Line 349-350 “Both the increase in mobility and decrease in lattice thermal conductivity with higher dyscrasite content could be explained by a decrease of grain boundaries scattering.” This statement is confusing; with the increase in dyscrasite content the number of grains is supposed to increase, enhancing the grain boundary scattering."

We would like to thank the reviewer for pointing out that mistake in our discussion; the experimentally observed increase in mobility can indeed not be explained by a decrease in grain boundary scattering. A possible reason is that the higher dyscrasite content leads to a higher effective mobility in the material.

To correct our discussion we have rewritten the paragraph starting at l359 as follows:

“The decrease in lattice thermal conductivity with higher dyscrasite content can be explained by a increase of grain boundaries scattering and/or defect scattering. Liu et al. find contradicting influence of the impurity content on the lattice thermal conductivity [2], which might be rather due to the presence of a second impurity (Ag3Li10), as our results are in agreement with two publications from Zheng et al. who explain the decrease in lattice thermal conductivity by the combination of defect scattering and the grain boundaries scattering. The observed increase in hole mobility is somewhat surprising; it could partially be explained by an effective medium effect due to the metallic nature of dyscrasite. The increase of carrier concentration and mobility and the decrease in lattice thermal conductivity result in an increase in , which shows that dyscrasite might not have a strong detrimental effect. 

Points 4 and 5 (dynamic links) have been corrected in the manuscript.

[1]    Liu, Z., et al, High thermoelectric performance of a-MgAgSb for power generation, 10.1039/C7EE02504A (Perspective) Energy Environ. Sci., 2018, 11, 23-44

[2]     Liu, Z., et al., Lithium doping to enhance thermoelectric performance of MgAgSb with weak electron–phonon coupling. Advanced Energy Materials, 2016. 6(7): p. 1502269.